# Comparative Antimicrobial Activity of Hp404 Peptide and Its Analogs against *Acinetobacter baumannii*

**DOI:** 10.3390/ijms22115540

**Published:** 2021-05-24

**Authors:** Min Ji Hong, Min Kyung Kim, Yoonkyung Park

**Affiliations:** 1Department of Biomedical Sciences, Chosun University, Gwangju 61452, Korea; mjhong94@naver.com (M.J.H.); charm5964@naver.com (M.K.K.); 2Research Center for Proteineous Materials, Chosun University, Gwangju 61452, Korea

**Keywords:** *Acinetobacter baumannii*, antimicrobial peptide, biofilm

## Abstract

An amphipathic α-helical peptide, Hp1404, was isolated from the venomous gland of the scorpion *Heterometrus petersii*. Hp1404 exhibits antimicrobial activity against methicillin-resistant *Staphylococcus aureus* but is cytotoxic. In this study, we designed antimicrobial peptides by substituting amino acids at the 14 C-terminal residues of Hp1404 to reduce toxicity and improve antibacterial activity. The analog peptides, which had an amphipathic α-helical structure, were active against gram-positive and gram-negative bacteria, particularly multidrug-resistant *Acinetobacter baumannii*, and showed lower cytotoxicity than Hp1404. *N*-phenyl-1-naphthylamine uptake and DisC_3_-5 assays demonstrated that the peptides kill bacteria by effectively permeating the outer and cytoplasmic membranes. Additionally, the analog peptides inhibited biofilm formation largely than Hp1404 at low concentrations. These results suggest that the analog peptides of Hp1404 can be used as therapeutic agents against *A. baumannii* infection.

## 1. Introduction

*Acinetobacter baumannii* is a rod-shaped gram-negative bacterium and among the major causes of opportunistic nosocomial infections, known as ESKAPE pathogens (*Enterococcus faecium*, *Staphylococcus aureus*, *Klebsiella pneumoniae*, *A. baumannii*, *Pseudomonas aeruginosa* and *Enterobacter* species). ESKAPE pathogens are highly antibiotic-resistant and difficult to treat, accounting for the majority of nosocomial and community-acquired infections and show high mortality rates [1,2]. The World Health Organization recently cited antibacterial resistance as one of the three most important problems facing human health and reported that the number of multidrug-resistant (MDR) *A. baumannii* infections has increased rapidly, posing a public health threat worldwide [3,4,5,6]. To treat *A. baumannii* infection, antibiotics such as β-lactams or carbapenems with a broad spectrum are used as first-line options, but strains that have recently gained carbapenem resistance have emerged. Polymyxin or tigecycline is used to treat carbapenem-resistant *A. baumannii* [7], but most strains are multidrug resistant, and thus treatment options are very limited [8]. Multidrug-resistant *A. baumannii* causes opportunistic infections, including septicemia, pneumonia, endocarditis, skin and wound infections and urinary tract infections following hospitalization of patients with severe illness [9,10]. This increased bacterial resistance to antibiotics is attributed to *A. baumannii*’s ability to form biofilms.

Biofilms are communities of microorganisms attached to a solid surface and produce a matrix known as extracellular polymeric substances (EPS) [11]. EPS are composed mostly of polysaccharides, glycoproteins, glycolipids and extracellular DNA [12,13,14,15]. The matrix acts to protect bacteria, enabling them to survive in difficult-to-grow environments. Therefore, enhanced resistance or resistance to antibiotics and other antimicrobial agents compared to plankton bacteria is a typical feature of biofilms. Most bacterial infections are thought to be related to bacterial biofilms, and many cases of device-related infections have been reported. Device-related infections typically occur in medical devices such as catheters, contact lenses, mechanical heart valves and artificial joints [16]. Therefore, the development of new antimicrobial drugs against these multidrug-resistant pathogens is needed; antimicrobial peptides (AMPs) are considered an alternative to antibiotics.

Organisms such as humans, animals and plants are exposed to various microbial at-tacks; they protect themselves from these threats by producing antimicrobial peptides (AMPs) via the innate immune system [17,18]. AMPs, which have cationic and amphipathic properties, consist of fewer than 50 amino acids and show potent activity against bacteria, fungi, yeast and viruses [19,20,21]. AMPs also exert anticancer activities and regulate inflammatory responses and wound healing [22,23,24]. Killing of bacteria leads to membrane destruction by causing the formation of pores or micelles and inhibition of protein synthesis or DNA replication by binding to RNA and DNA [25]. Despite the outstanding properties of AMPs, they exhibit disadvantages such as hemolytic activity against red blood cells and sensitivity to salt. Since antimicrobial peptides have the disadvantage of being sensitive to a salt environment, it is important to develop peptides that maintain their antibacterial activity in the presence of salts [26]. AMPs that can overcome these shortcomings would be promising candidates for treating bacterial infectious diseases [27]. 

In a previous study, the new amphipathic peptide Hp1404 was identified from the venomous gland cDNA library of the scorpion *Heterometrus petersii* [28]. Hp1404 consists of 14 amino acids, has a net charge of + 1 and exhibits an α-helical structure in a bacterial membrane-mimicking environment. Hp1404 has been reported to have potential antibac-terial activity, including against only gram-positive bacteria, such as Staphylococcus aureus. However, Hp1404 has a relatively cytotoxicity in mammalian cells and a high level of hemolysis of red blood cells.

Hence, the purpose of this study has been to design a peptide less toxic than its parent Hp1404 peptide and to investigate the antibacterial activity of the newly designed antimicrobial peptide on multidrug-resistant strains. We designed analog peptides by replacing the position 14 amino acid with different amino acids. Antimicrobial activity against gram-positive, gram-negative and multidrug-resistant bacteria of the designed analog peptides was tested, and their mechanism of action was examined. Furthermore, we investigated the activity of peptides against *A. baumannii* strains and their ability to inhibit and eliminate bacterial biofilms. Our findings suggest that the newly designed peptides have the potential to be promising candidates as antimicrobial agents against *A. baumannii* strains.

## 2. Results

### 2.1. Peptide Design and Characterization 

The analog peptides were designed based on the peptide Hp1404 by substituting the F residue with six different amino acids (A, K, V, L, I and W). Wheel diagram analysis showed that hydrophobic and hydrophilic amino acids formed an amphipathic structure (Figure 1). The peptide sequence, retention time, molecular weight, hydrophobicity and net charge of Hp1404 and its analog peptides are listed in Table 1. The hydrophobicity values of Hp1404, Hp1404-A, Hp1404-K, Hp1404-V, Hp1404-L, Hp1404-I and Hp1404-W were 0.686, 0.580, 0.487, 0.645, 0.679, 0.686 and 0.719, respectively. The parent peptide and five analog peptides showed a net charge of +1, whereas Hp1404-K had a net charge of +2.

### 2.2. Secondary Structure of the Peptides

Circular dichroism (CD) spectroscopy was performed to determine the secondary structure of the peptides in various environments. This analysis was performed in 10 mM SP buffer (aqueous environment), bacterial membrane-mimicking solvents such as 30 mM sodium dodecyl sulfate (SDS; negatively charged environment) and 50% 2, 2, 2-trifluoroethanol (TFE; hydrophobic environment). As shown in Figure 2, in the aqueous environment, all peptides displayed a random coil conformation, as indicated by the minimum peak at 198 nm. In the presence of 30 mM SDS and 50% TFE, the parent peptide and all analog peptides had an α-helical conformation, as demonstrated by minimum peaks at 208 and 222 nm. These results showed that the analog peptides affected the bacterial membrane by forming an α-helix structure.

### 2.3. Antimicrobial Activity

The antimicrobial activity of Hp1404 and its analog peptides against gram-positive and gram-negative bacteria was determined using the two-fold broth microdilution method. The results are summarized in Table 2. The parent peptide, Hp1404, displayed minimum inhibitory concentration at 3.13 ranging from 12.5 μM against gram-positive and gram-negative bacteria. All four analog peptides (except Hp1404-A and Hp1404-K) showed broad-spectrum antimicrobial activities, whereas Hp1404-A and Hp1404-K exhibited weak effects towards only a few of these tested strains. Hp1404-L and Hp1404-I exhibited a minimum inhibitory concentration (MIC) of 3.13 μM against *A. baumannii.*

In addition, these analog peptides were effective against multidrug-resistant *A. baumannii* strains, with Hp1404-L and Hp1404-W showing the same activity as the parent peptide (Table 3). Thus, the analog peptides have excellent antibacterial activity against *A. baumannii*.

### 2.4. Cytotoxicity Assay

We evaluated the hemolytic activity of the peptides against mouse erythrocytes. All peptides with amino acid substitutions showed lower hemolysis than Hp1404 (Figure 3a). At the highest tested concentration of 50 μM, Hp1404, Hp1404-L and Hp1404-W showed hemolytic activities of 49.2%, 45.8% and 15.9%, respectively. However, Hp1404-A, Hp1404-K and Hp1404-V showed no hemolytic activity. The parent peptide Hp1404 showed hemolytic activity of 41.7%, similar to the rate at 50 μM, whereas Hp1404-L, Hp1404-I and Hp1404-W induced 21.3%, 0% and 1.7% hemolysis at 25 μM. Melittin, used as a positive control, showed 42% hemolytic activity at concentrations up to 1.56 μM, indicating that it is toxic.

To confirm the cytotoxicity of Hp1404 and its analog peptides toward human keratinocyte HaCaT cells, treatment with various peptide concentrations after 24 h was measured by MTT assay. Similar to the hemolytic activity, cell viability was lowest after treatment with Hp1404 and Hp1404-L with values of 72.3% and 62.4%, respectively, against HaCaT cells at 50 μM. Cell viability was 100% after treatment with Hp1404-A, Hp1404-V, Hp1404-I and Hp1404-W (Figure 3b). Compared to melittin, analog peptides showed significantly lower toxicity. Cell viability was high after treatment with all analog peptides, indicating that they were not cytotoxic.

### 2.5. Calcein Leakage Assay

As shown in Figure 4a,b, to evaluate the toxicity of the peptides via their effects on the cell membranes, large unilamellar vesicles (LUVs) containing calcein were used in a leakage assay. Upon treatment with various concentrations of peptides at LUVs, calcein was released in a concentration-dependent manner. For Hp1404-A and Hp1404-K, phosphatidylcholine: cholesterol (PC:CH) liposomes (mimicking human erythrocyte membrane) and PC:CH: sphingomyelin (SM) liposomes (mimicking mammalian cell membrane) did not release calcein, indicating no toxicity. Following treatment with melittin, which was used as a positive control, regardless of the concentration, the fluorescence intensity of calcein was released up to 1500 arbitrary units (a.u.). Hp1404-V, Hp1404-I, Hp1404-L and Hp1404-W caused similar calcein leakage as the parent peptide Hp1404 and was less toxic than melittin. Furthermore, calcein leakage was assessed using a fluorescence-labeled organism bioimaging instrument. As shown in Figure 4c, the analog peptides (Hp1404-V, Hp1404-I, Hp1404-L and Hp1404-W) caused lower leakage activity in liposomes compared to the positive control melittin and parent peptide Hp1404, and no fluorescence was observed for Hp1404-A and Hp1404-K. Based on these results, the analog peptides are not highly toxic towards mammalian cell membranes.

### 2.6. Salt Stability

To evaluate the effect of a salt-containing environment on the antibacterial peptide, peptides were assessed by MIC assay in the presence of 150 mM NaCl. As shown in Figure 5, the MIC values of Hp1404, Hp1404-L and Hp1404-W for *A. baumannii* KCTC 2508 increased from 3.13 to 6.25 μM and Hp1404-I and Hp1404-V increased from 3.13 to 12.5 μM and 6.25 to 25 μM, respectively, reducing the antibacterial activity. In all resistant strains, Hp1404 and analog peptides has antibacterial activity at 4-fold the MIC value.

### 2.7. Time-Killing Assay

To measure bacterial mortality over time caused by the peptides, we performed time-kill analysis. The peptides, except for Hp1404-I and Hp1404-V at 1x MIC, quickly killed the bacteria within 60 min. However, at 4× MIC, analog peptides killed bacteria within 10 min, similar to the parent peptide Hp1404. Based on these results, both Hp1404 and its analog peptides killed bacteria in a short time (Figure 6).

### 2.8. Biofilm Inhibition

We quantified biofilm formation in a crystal violet staining assay. The absorbance values of the standard strain and six clinically isolated strains were measured, and the highest formation rate of the four strains was used to confirm biofilm inhibition ability. As shown in Figure 7a, Hp1404 and its 4 analog peptides (Hp1404-V, Hp1404-L, Hp1404-I and Hp1404-W) inhibited *A. baumannii* KCTC 2508 biofilm formation at 6.25, 12.5, 6.25, 6.25 and 6.25 μM, respectively. The antibiotics meropenem and polymyxin B also inhibited biofilm formation at 6.25 μM. For *A. baumannii* #3 and 409081, Hp1404 inhibited biofilm formation at 6.25 μM and its 4 analog peptides inhibited this formation at 12.5 μM. Meropenem did not affect biofilm inhibition at less than 25 μM, and polymyxin B showed inhibition at 6.25 μM (Figure 7b,d). In *A. baumannii* #4, Hp1404, Hp1404-V, Hp1404-L, Hp1404-I and Hp1404-W inhibited biofilm formation at 6.25, 12.5, 6.25, 12.5 and 12.5 μM, respectively. Additionally, Hp1404, Hp1404-V, Hp1404-L, Hp1404-I, and Hp1404-W showed biofilm inhibition effects at 6.25, 12.5, 6.25, 12.5 and 12.5 μM, respectively, in *A. baumannii* 719705. Similarly, for *A. baumannii* #3 and 719705, meropenem exhibited no biofilm inhibition activity and polymyxin B showed activity at 6.25 μM (Figure 7c,e).

### 2.9. Biofilm Reduction and Visualization

We performed a crystal violet staining assay to evaluate the biofilm reduction ability of the peptides and antibiotics (Figure 8a–c). In *A. baumannii* KCTC 2508, Hp1404, Hp1404-V, Hp1404-L, Hp1404-I and Hp1404-W showed reduced biofilm activity at 50 μM, with reductions of 74.06%, 79.50%, 84.72%, 83.82% and 79.90%, respectively. Hp1404 and its analog peptides (Hp1404-V, Hp1404-L, Hp1404-I and Hp1404-W) reduced biofilm formation by up to 80% in the baumannii #3 at 100, 50, 50, 50 and 50 μM, respectively. For *A. baumannii* 409081, Hp1404, Hp1404-V, Hp1404-L, Hp1404-I and Hp1404-W peptides of 50 μM reduced biofilm formation by 34.52%, 53.23%, 64.76%, 72.68% and 65.08%, respectively. All analog peptides removed the biofilm at a lower concentration compared to Hp1404. Polymyxin B showed lower MIC values than the peptides and strong biofilm inhibition effects, but the ability to remove formed biofilm was low.

The biofilm reduction ability was confirmed by visualizing *A. baumannii* biofilms using an EVOS™ FL Auto 2 imaging system (Figure 8d). The biofilm was treated with 50 μM Hp1404, its analog peptides and meropenem or polymyxin B. Treatment with Hp1404-I and Hp1404-W, decreased SYTO9 fluorescence by more than Hp1404 in all strains. In contrast, meropenem or polymyxin B did not decrease biofilm formation of *A. baumannii*, including the strains isolated from patients.

### 2.10. EPS Determination

EPS are a complex mixture of carbohydrates, DNA and proteins in biofilms. We measured the total carbohydrate content in EPS using the phenol-sulfuric acid method. After the formation of biofilm, treatment with 50 μM peptides or antibiotics resulted in reduced EPS production (Figure 9). For the standard strain *A. baumannii* KCTC 2508, the control showed 0.4 μg/mL EPS production. After treatment with the peptides, EPS production decreased from 0.1 to 0.25 μg/mL. Antibiotics reduced this value from 0.12 to 0.15 μg/mL, which is less than that after peptide treatment. Antibiotic-resistant strains also reduced EPS production. Thus, the peptides and antibiotics caused similar reductions in EPS production.

### 2.11. NPN Uptake

The increase in fluorescence due to *N*-phenyl-1-naphthylamine (NPN) in the outer membrane was measured to determine the outer membrane permeability of the peptides. NPN fluorescence increases when the bacterial outer membrane is disrupted, and hydrophobic regions are exposed by the action of the peptide. As shown in Figure 10, Hp1404 and its analog peptides induced NPN fluorescence in *A. baumannii* cells in time -and concentration-dependent manners. The fluorescence intensity of 4× MIC Hp1404 and its analog peptides (Hp1404-V, Hp1404-L, Hp1404-I and Hp1404-W) for 30 min reached approximately 600, 650, 540, 530 and 490 arbitrary units (a.u.), respectively.

### 2.12. DisC_3_-5

We assessed whether Hp1404 and its analog peptides induced membrane depolarization using DisC_3_-5, which accumulates in intact cytoplasmic membrane cells and fluoresces when the membrane is disrupted by the peptides. All peptides produced time- and dose-dependent effects (Figure 11). Hp1404 and four analog peptides (Hp1404-V, Hp1404-L, Hp1404-I and Hp1404-W) at 4× MIC increased the fluorescence intensity to almost 210, 200, 200, 220 and 260 arbitrary units (a.u.), respectively. Analog peptides induced cytoplasmic membrane depolarization comparable to that of Hp1404. DisC_3_-5 fluorescence increased immediately to the highest level after addition of Hp1404. Hp1404 and its analog peptides caused rapid membrane depolarization, and Hp1404-W showed higher fluorescence over time.

## 3. Discussion

The venom of various animal species is the source of biologically active molecules with attractive candidates for the development of novel therapeutics [29]. Antimicrobial peptides (AMPs) are bioactive molecules and notable sources of known antibacterial activity found in scorpions, ants, bees, snakes, wasps and spiders. AMPs isolated from scorpions have been reported to effectively kill multidrug-resistant pathogens and possess anticancer activity but are slightly toxic [30]. Modifying AMPs could improve the antibacterial effect while reducing cytotoxicity, thereby overcoming the barriers preventing their development as therapeutic agents. Thus, new peptide analogs of *Heterometrus petersii* scorpion AMPs have been designed and synthesized, and their antimicrobial spectrum and hemolytic/cytotoxic activity have been determined.

In the current study, we predicted that by replacing various amino acids in Hp1404, the peptides would show increased hydrophobicity and reduced toxicity. Six analog peptides of Hp1404 were designed by replacing phenylalanine (Phe, F) with an electrically charged amino acid, lysine (Lys, K) [31] and hydrophobic amino acids, alanine (Ala, A) [32], valine (Val, V) [33], leucine (Leu, L) [34], isoleucine (Ile, I) [35] and tryptophan (Trp, W) [36]. The net charge was increased by substituting the charged amino acid lysine, and hydrophobicity was similar to or increased by substitution with hydrophobic amino acids.

We examined the antimicrobial activity of the parent peptide Hp1404 and its six analog peptides against gram-positive and gram-negative bacteria. Hp1404 and four analog peptides (Hp1404-W, Hp1404-L, Hp1404-I and Hp1404-W) displayed antimicrobial activity, with excellent activity against *A. baumannii*, including clinical strains resistant to antibiotics. Particularly, the peptides substituted with Lue and Trp maintained antibacterial activity. Hp1404 and melittin were toxic towards mammalian cells, but the toxicity of the analog peptides decreased and cell viability increased by 80% even at a concentration 4-fold the active concentration, and the analogs exhibited stable antibacterial activity in a high-salt environment. Hp1404-L maintained antibacterial activity but showed toxicity similar to that of the parent Hp1404 peptide, while Hp1404-W showed no toxicity. For use as therapeutic agents, peptides need to have low toxicity and the ability to maintain the activity in a salt environment [37]. Cations such as Na^+^ interfere with the electrostatic interaction between the peptide and bacterial membrane, thereby reducing antimicrobial activity. Therefore, we determined whether each peptide maintains antibacterial activity in the presence of Na^+^; Hp1404-L and Hp1404-W showed excellent activity along with the parent peptide Hp1404. Thus, the analog peptides show potential for treating *A. baumannii* infections and overcome the limitations of Hp1404.

AMPs display random coil structures in their native forms; in bacterial membrane-mimicking environments, they display α-helix, β-sheet or loop structures which act on the bacterial membrane [38]. It is known that AMPs such as melittin [39], magainin2 [40], cecropins [41] and LL37 [42] adopt α-helix structures. Hp1404 has been reported to form amphipathic α-helix when adsorbed to the bilayer lipid membrane of the bacterial membrane [28]. The structures of the analog peptides were confirmed under different conditions by CD spectroscopy. As a result, the parent peptide and its analog peptides formed a random coil in aqueous solution, and an α-helical structure was adopted in membrane-mimicking environments. The high amphiphilic and strong helix properties of the peptides are associated with their antimicrobial activity in high-salt environments (Figure 5). The helix structure and positive charges are strongly bound to negatively charged phospholipid bacterial membranes and efficiently exert their effects on them. We investigated the mechanism of action of outer membrane permeability and cytoplasmic membrane depolarization using NPN and DisC_3_-5. Peptides showed an increased fluorescence intensity of NPN and DiSC_3_-5 at each active concentration. This is consistent with the findings of studies on the mechanisms of action of AMP such as LL37 and melittin that act by destroying bacterial membranes. These results suggest that the peptides interact with the bacterial membrane to exert antibacterial effects by causing outer and inner membrane permeability in *A. baumannii*. Therefore, we observed that owing to this mechanism of action, analog peptides killed the bacteria in short time and thus showed potential as therapeutic agents.

Biofilm formation increases antibiotic resistance compared to strains in the planktonic state, and there are concerns that the effect of the peptide may be reduced by the production of polysaccharides. Microorganisms attach to surfaces such as medical devices and produce a polymer matrix containing extracellular DNA, polysaccharides and glycoproteins that cause infection [43,44]. The tolerance of these biofilms is increased by endogenous oxidative stress, upregulation of the effluent pump and common tolerance mechanisms such as beta-lactamase [45]. To assess biofilm inhibition activity, we used Hp1404, four analog peptides and two conventional antibiotics. The parent and its analog peptides showed similar levels of inhibition of biofilm formation. Meropenem inhibited biofilm formation of the standard strain *A. baumannii* KCTC 2508 but did not inhibit that of drug-resistant *A. baumannii* strains. Polymyxin B inhibited biofilm formation of *A. baumannii*, including that of resistant strains. For biofilm eradication, analog peptides were more effective than Hp1404, with Hp1404-I showing the strongest effects. Polymyxin B was superior for inhibiting biofilm formation but exhibited a significantly lower eradication ability than the peptides. Furthermore, the effect of the peptides on EPS secreted by the biofilm was confirmed. Analog peptides reduced EPS production to a level similar as that of Hp1404. Thus, analog peptides of Hp1404 may be useful alternatives to antibiotics.

In summary, we designed Hp1404 analogs with reduced toxicity by substituting the amino acid in position 14. The analog peptides showed activity against gram-positive and gram-negative bacteria. We determined the activity of these peptides against multidrug-resistant *A. baumannii* and their anti-biofilm effects. The toxicity of the analog peptides was lower, and the anti-biofilm effect was better than that of Hp1404. Thus, these newly designed peptides are promising candidates as antimicrobial agents against *A. baumannii* strains.

## 4. Materials and Methods

### 4.1. Materials

SDS, TFE, thiazolyl blue tetrazolium bromide (MTT), NPN, 3,3′-dipropylthiadicarbocyanine iodide (DisC_3_-5), SYTOX green, propidium iodide (PI), dimethyl sulfoxide, meropenem and polymyxin B were purchased from Sigma-Aldrich (St. Louis, MO, USA). Vancomycin was purchased from LPS solution (Daejeon, South Korea).

### 4.2. Microorganisms

*Staphylococcus aureus* ATCC 25923, S. aureus ATCC 29213, *P. aeruginosa* ATCC 27853, *Escherichia coli* ATCC 25922 and *Salmonella typhimurium* ATCC 14028 strains were obtained from the American Type Culture Collection (ATCC, Manassas, VA, USA). *Listeria monocytogenes* KCTC 3710, *Bacillus cereus* KCTC 1012, K. pneumoniae KCTC 2208 and *A. baumannii* KCTC 2508 were obtained from the Korean Collection for Type Cultures (KRIBB, Daejeon, South Korea). Clinical isolates of *A. baumannii* strains were collected from Eulji University Hospital (Seoul, South Korea).

### 4.3. Peptide Design and Characterization

The analog peptides were designed by substituting amino acids in Hp1404. The three-dimensional structures of Hp1404 and its analog peptides were constructed using Mobyle@RPBS (http://mobyle.rpbs.univ-paris-diderot.fr/cgi-bin/portal.py#welcome accessed on 14 August 2020). Helical wheel diagrams and the hydrophobicity of the peptides were obtained using the HeliQuest site (http://heliquest.ipmc.cnrs.fr, accessed on 14 August 2020).

### 4.4. Peptide Synthesis

The peptides were synthesized using the solid-phase-9-fluorenylmethoxycarbonyl (Fmoc) method, as reported previously [46], on a Rink amide 4-methylbenzhydrylamine resin using a Liberty microwave peptide synthesizer (CEM, Matthews, NY, USA). The following chemicals were used as linkage reagents: 0.45 M 2-(1H-benzotriazole-1-yil)-1,1,3,3-tetramethyluronium hexafluorophosphate (HBTU) diluted in dimethylformamide (DMF), 0.1 M *N*-hydroxybenzotriazole (HOBt) diluted in piperidine/DMF and 2 M *N**, N*-diisopropylethylamine (DIEA) diluted in *N*-methylpyrrolidone (NMP). After washing with dichloromethane (DCM), cleavage was performed by incubating for 2 h at 25 °C in a trifluoroacetic acid (TFA) solution containing water, phenol and triisopropylsilane. The crude peptide was precipitated by dilution with ice-cold diethyl ether, and then spread on the tube wall and dried. After resuspension in water at 25 °C, the peptide was purified by reversed-phase high-performance liquid chromatography (RP-HPLC) on a Jupiter C18 column (4.6 × 250 mm, 300 Å, 5 µm; Phenomenex, Torrance, CA, USA). The molecular weights of the peptides were confirmed using matrix-assisted laser desorption ionization-time of flight (MALDI-TOF) mass spectrometry (Kratos Analytical Inc., Chestnut Ridge, NY, USA). The peptides were dissolved in deionized water (DI H_2_O) and stored at −20 °C. In this study, all the peptides used were >95% pure.

### 4.5. CD Spectroscopy

CD spectra were measured to determine the secondary structure of the peptide, which was changed by the bacterial membrane-mimicking environment. The peptides were used at a concentration of 40 μM. The secondary structure of the peptides was investigated in 10 mM SP buffer (pH 7.2), 30 mM SDS and 50% TFE. Spectra were investigated at wavelengths ranging from 190 to 250 nm using a quartz cell with a 1.0-mm path length on a JASCO 810 spectropolarimeter (Jasco, Tokyo, Japan) [47].

### 4.6. Antibacterial Activity Test

Hp1404 and its analog peptides activity against both gram-positive and gram-negative bacterial strains were tested using the broth microdilution method [48]. Briefly, S. aureus (ATCC 25923, ATCC 29213) and B. cereus KCTC 1012 were cultured overnight at 37 °C in tryptic soy broth (TSB). P. aeruginosa ATCC 27853, K. pneumoniae KCTC 2208, *A. baumannii* KCTC 2508 and S. typhimurium ATCC 14028 were cultured overnight at 37 °C in nutrient broth (NB). E. coli ATCC 25922 was cultured overnight at 37 °C in luria bertani (LB). L. monocytogenes KCTC 3710 was cultured overnight at 37 °C in brain heart infusion (BHI). The cultures were diluted with fresh medium at a density of 2 × 105 CFU/mL. Peptides were serially diluted in sodium phosphate buffer (pH 7.2) to a final concentration of 25 μM. In a 96-well plate, 50-μL bacterial aliquots were mixed with 50 μL peptide solution. After incubation at 37 °C for 18 h, the MIC was determined using a plate reader at 600 nm. The experiment was performed in triplicate.

### 4.7. Salt Stability Assay

To determine whether antibacterial activity was affected by the presence of salts, experiments were conducted as previously reported [49]. We determined the MICs of NB supplemented with 150 mM NaCl. MIC assays were performed as described above. The assay was performed in triplicate.

### 4.8. Hemolytic Assay

The hemolytic activity of the peptides against mouse erythrocytes was determined using red blood cells [50]. Fresh mouse red blood cells were washed three times and resuspended in PBS to attain an 8% dilution. Peptides were serially diluted to a final concentration of 50 μM. Next, 100 µL peptide solution was added to 100 μL of mouse red blood cells, wrapped with aluminum foil and incubated for 1 h at 37 °C. The mixture was then centrifuged at 1500× *g* for 10 min. The supernatant was transferred into a new 96-well plate, and the absorbance of the supernatant was measured at 414 nm. As positive and negative controls, PBS and 0.1% Triton X-100 in mouse red blood cells were used. Hemolysis was evaluated according to the following equation:

Hemolysis (%) = [(Abs414 nm in the peptide solution − Abs414 nm in PBS)/(Abs414 nm in 0.1% Triton X-100 − Abs414 nm in PBS)] × 100.

### 4.9. Cytotoxicity Assay

The cytotoxicity of the peptides against mammalian cells was calculated using the MTT assay. Human keratinocyte HaCaT cells were incubated in Dulbecco’s modified Eagle’s medium (DMEM) supplemented with 10% fetal bovine serum and 1% penicillin at 37 °C with 5% CO_2_. HaCaT cells were added at 2 × 105 cells/mL to DMEM supplemented with 10% fetal bovine serum and placed in 96-well plates. The plates were incubated overnight at 37 °C in a CO_2_ incubator. Peptides were serially diluted two-fold in DMEM, added to the cell cultures and incubated for 24 h at 37 °C. MTT solution (5 mg/mL) was added to each well and incubated for 1 h at 37 °C. The media was discarded, and the formazan crystals were dissolved in 100 μL dimethyl sulfoxide. The absorbance was measured using a Versa-Max microplate ELISA reader (Molecular Devices, Sunnyvale, CA, USA) at 570 and 650 nm. As a negative control, 0.1% Triton X-100 was used.

### 4.10. Time-Killing Assay

A time-killing assay of the peptides against *A. baumannii* KCTC 2508 was performed. The bacteria were grown to mid-log phase in NB medium, with cell density of *A. baumannii* KCTC 2508 of 2 × 10^5^ CFU/mL. The peptides were used to treat the bacteria at concentrations of 1×, 2× and 4× MIC and incubated at 37 °C with shaking for 0–180 min. The reactive aliquots were plated on NB agar and incubated at 37 °C for 24 h, and surviving bacterial cells were counted. The assay was conducted in triplicate.

### 4.11. Calcein Leakage Assay

Peptide-induced membrane permeability was monitored be measuring calcein dye emission in calcein-loaded LUVs [49]. We prepared the lipid mixture, and PC:CH (2:1, *w*/*w*) and PC:CH:SM liposomes (1:1:1, *w*/*w*) were prepared with 70 mM calcein buffer. A Sephadex G-50 column was used to separate calcein-loaded liposomes from free calcein dye. The LUVs (10 μM) were mixed with various concentrations of the peptide on a 96-well black plate. The fluorescence intensity was measured at excitation and emission wavelengths of 480 and 520 nm, respectively.

### 4.12. Biofilm Inhibition Assay

Acinetobacter baumannii KCTC 2508, #3, #4, 409081 and 719705 were incubated overnight at 37°C to mid-log phase in NB medium. Bacterial cells (5 × 10^5^ CFU/mL) were suspended in NB containing 0.2% glucose. A final concentration of 25 μM of peptides and antibiotics (meropenem and polymyxin B) was prepared with the bacterial suspensions in 96-well tissue culture plates and incubated at 37 °C for 24 h. The culture media was carefully removed, and biofilms were fixed with 100% methanol for 15 min and then stained with 0.1% crystal violet for 1 h and rinsed three times with distilled water. After air-drying, the biofilm mass was dissolved in 100 μL of 95% ethanol and measured at 595 nm using a microplate reader [51].

### 4.13. Biofilm Reduction Assay

Acinetobacter baumannii KCTC 2508, #3 and 409081 cells were cultured in NB at 37 °C. Bacterial cells were diluted to 5 × 10^5^ CFU/mL in fresh NB with 0.2% glucose in a 96-well plate and incubated for 24 h. After incubation, the wells were washed with PBS to remove non-adherent cells, and serially diluted peptides and antibiotics (final concentration 200 μM) were treated for 24 h. A crystal violet assay was performed, and biofilm formation was measured as described above.

### 4.14. Live Biofilm Viability Assay

An *A. baumannii* biofilm viability assay was performed by staining with Syto9 dye [52]. Acinetobacter baumannii strains were diluted to approximately 5 × 10^5^ CFU/mL and incubated for 24 h. Biofilms of *A. baumannii* strains were washed with PBS and treated with 25 and 50 μM peptides and antibiotics for 24 h. Biofilms were stained with SYTO 9 diluted in PBS and the biofilm was visualized using an EVOS™ FL Auto 2 Imaging System (Thermo Fisher Scientific, Waltham, MA, USA).

### 4.15. EPS Determination

The total carbohydrate content in EPS was measured using the phenol-sulfuric acid method with glucose as a standard [53]. Briefly, biofilms of *A. baumannii* strains were incubated for 24 h, washed with PBS and treated with 50 μM peptides or antibiotics. After 24 h, 100 μL EPS solution was mixed with 100 μL 5% phenol. Next, 500 μL sulfuric acid was added, and the mixture was incubated at room temperature for 1 h in the dark. The mixture was cooled on ice for 15 min and centrifuged 10,000× *g* for 10 min. The supernatant was transferred to a 96-well plate. Absorbance was read at 490 nm with a microplate reader.

### 4.16. NPN Uptake Assay

The hydrophobic fluorescent dye NPN was used to investigate whether the peptides induced outer membrane permeability [54,55]. Briefly, *A. baumannii* KCTC 2508 was grown in NB at 37 °C, washed three times and diluted to an OD600 of 0.2, in 5 mM HEPES buffer (pH 7.3). Diluted bacterial cells (100 μL) were added to 96-well black microplates and mixed with 50 μL of 10 μM NPN (dissolved in 95% ethanol), and background fluorescence was measured. Next, 0.5×, 1×, 2× and 4× MIC of peptides were added, and the increase in fluorescence caused by the peptides was recorded for 30 min. HEPES buffer (5 mM) was used as a negative control. The excitation and emission wavelengths were 350 and 420 nm, respectively.

### 4.17. Inner Membrane Depolarization Using DisC_3_-5

The inner membrane depolarization in the presence of peptides was determined using a membrane potential-dependent probe, DisC_3_-5. Briefly, mid-log phase *A. baumannii* KCTC 2508 cells were washed three times with 5 mM HEPES buffer (pH 7.3) containing 20 mM glucose. Next, 100 µL bacterial suspension (OD600 = 0.05) and 50 μL of a final concentration 1 μM DisC_3_-5 (in 5 mM HEPES with 20 mM glucose and 0.1 M KCl) mixtures were added to 96-well black microplates and incubated for 60 min until dye uptake reached a maximum. The peptides (0.5×, 1×, 2× and 4× MIC of peptides) were treated at 50 μL, and the change in fluorescence intensity was monitored for 30 min at excitation and emission wavelengths of 622 and 670 nm, respectively [56].

### 4.18. Statistical Analysis

All experiments were performed in triplicates in three independent experiments. Data are presented as the mean ± SEM. Statistical analysis of the results was performed with two-tailed Student’s t-test and one-way analysis of variance using GraphPad Prism version 8. 0 (La Jolla, CA, USA). The results were considered significant at * *p* < 0.05, ** *p* < 0.01, *** *p* < 0.001, **** *p* < 0.0001.

## 5. Conclusions

We designed analog peptides (Hp1404-V, Hp1404-L, Hp1404-I and Hp1404-W) that are less toxic than the parent peptide Hp1404 but have the same antibacterial activity against *A. baumannii* strains. Furthermore, the peptides maintained antibacterial activity under high-salt conditions and effectively inhibited biofilm formation and elimination. The analog peptides showed an α-helical structure in a bacterial membrane-mimicking environment and the same mechanism of action as the parent peptide. Overall, Hp1404-V, Hp1404-L, Hp1404-I and Hp1404-W are potential therapeutic agents against *A. baumannii* strains.

## Figures and Tables

**Figure 1 ijms-22-05540-f001:**
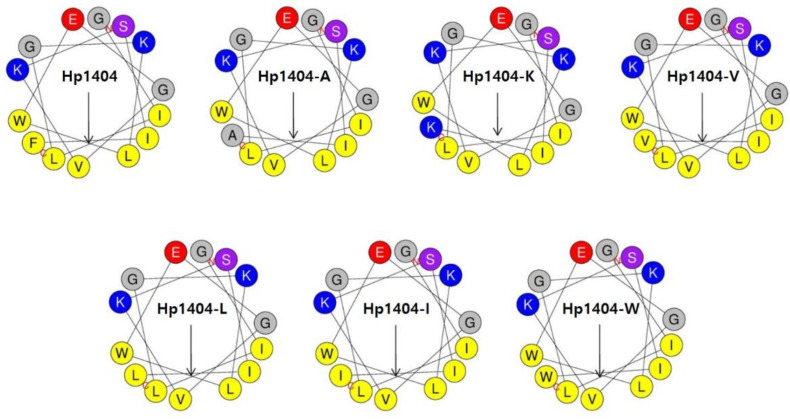
Wheel diagram of Hp1404 and its analog peptides. Wheel diagram was obtained from the HeliQuest site. Negatively charged residue is shown in red, positively charged residues are shown in blue, hydrophobic residues are shown in yellow and certain polar residues are shown in purple. Arrows indicate helical hydrophobic moments.

**Figure 2 ijms-22-05540-f002:**
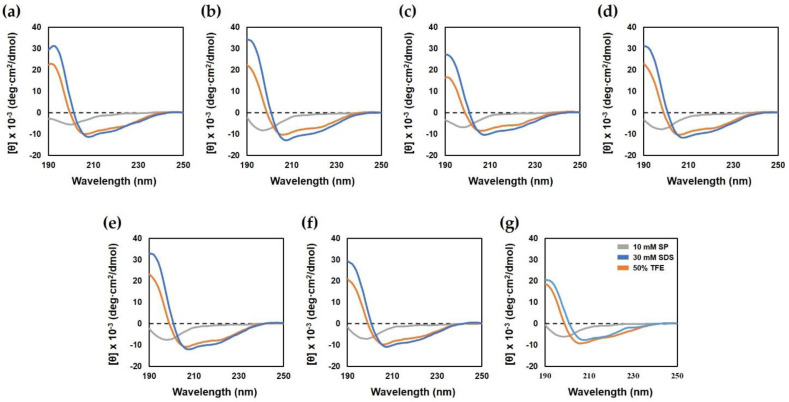
Circular dichroism spectra of Hp1404 and its analog peptides. The secondary structure of peptides was measured in 10 mM SP buffer, 50% TFE and 30 mM SDS by CD spectrometry. (**a**) Hp1404, (**b**) Hp1404-A, (**c**) Hp1404-K, (**d**) Hp1404-V, (**e**) Hp1404-L (**f**) Hp1404-I and (**g**) Hp1404-W peptide concentration was fixed at 40 μM.

**Figure 3 ijms-22-05540-f003:**
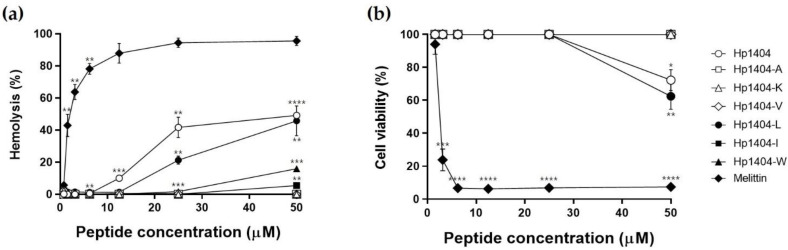
Hemolytic activity and cytotoxicity of peptides. (**a**) Hemolytic activity of peptides against 8% mouse red blood cells. (**b**) Cytotoxicity of peptides against HaCaT cells determined by MTT assay. Data are presented as the mean ± SEM of three independent experiments. * *p* < 0.05, ** *p* < 0.01, *** *p* < 0.001, **** *p* < 0.0001 versus the control group.

**Figure 4 ijms-22-05540-f004:**
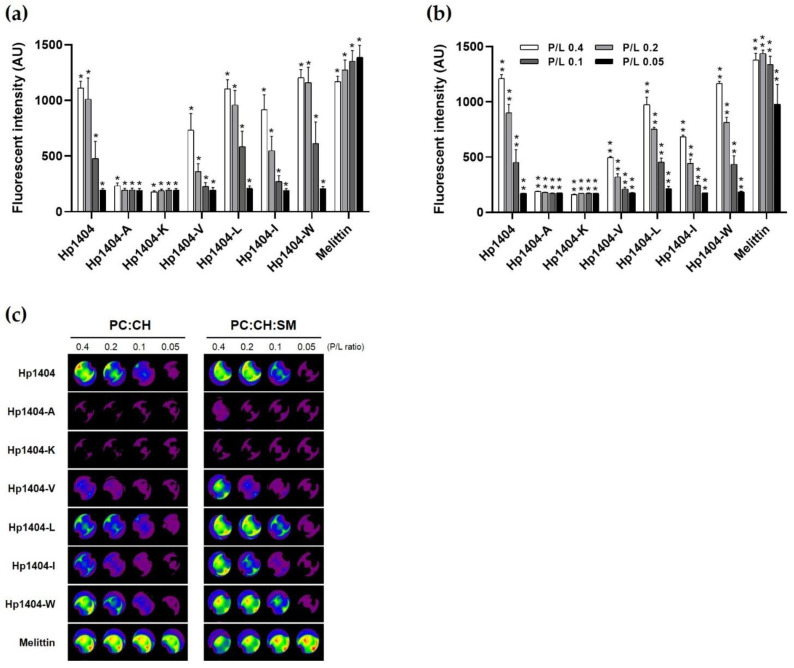
Calcein leakage assay. (**a**) PC:CH (1:1, *w*/*w*) and (**b**) PC:CH:SM (1:1:1, *w*/*w*) LUV emission intensity after treatment with the peptides. (**c**) Visualization of calcein release from calcein-loaded LUVs using a fluorescence-labeled organism bioimaging instrument. Data are presented as the mean ± SEM of three independent experiments using one-way ANOVA. * *p* < 0.05, ** *p* < 0.01.

**Figure 5 ijms-22-05540-f005:**
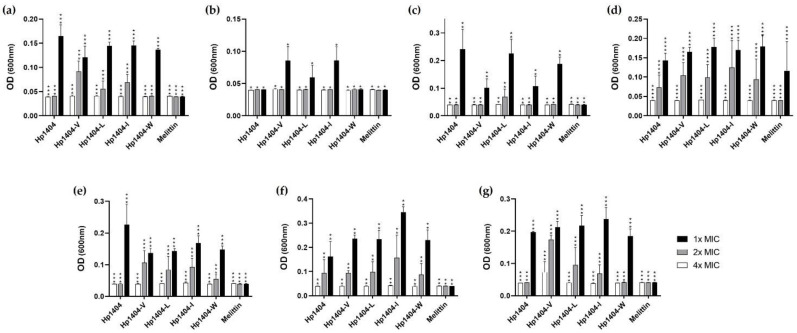
Antibacterial activity of peptides in salt environments. Salt stability of peptides were measured in the presence of 150 mM NaCl against *A. baumannii*. (**a**) *A. baumannii* KCTC 2508, (**b**) *A. baumannii* 409081, (**c**) *A. baumannii* 719705, (**d**) *A. baumannii* 907233, (**e**) *A. baumannii* #2, (**f**) *A. baumannii* #3 and (**g**) *A. baumannii* #4. Data are presented as the mean ± SEM of three independent experiments using one-way ANOVA. * *p* < 0.05, ** *p* < 0.01, *** *p* < 0.001, **** *p* < 0.0001.

**Figure 6 ijms-22-05540-f006:**
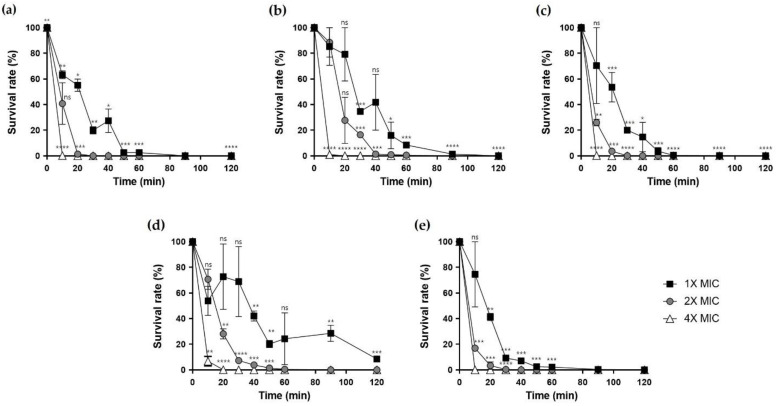
Time-killing assay. Peptides were prepared at concentrations of 1×, 2× and 4× MIC and treated against *A. baumannii* KCTC 2508. (**a**) Hp1404, (**b**) Hp1404-V, (**c**) Hp1404-L, (**d**) Hp1404-I and (**e**) Hp1404-W. Data are presented as the mean ± SEM of three independent experiments. * *p* < 0.05, ** *p* < 0.01, *** *p* < 0.001, **** *p* < 0.0001 versus the control group. ns = Not statistically significant.

**Figure 7 ijms-22-05540-f007:**
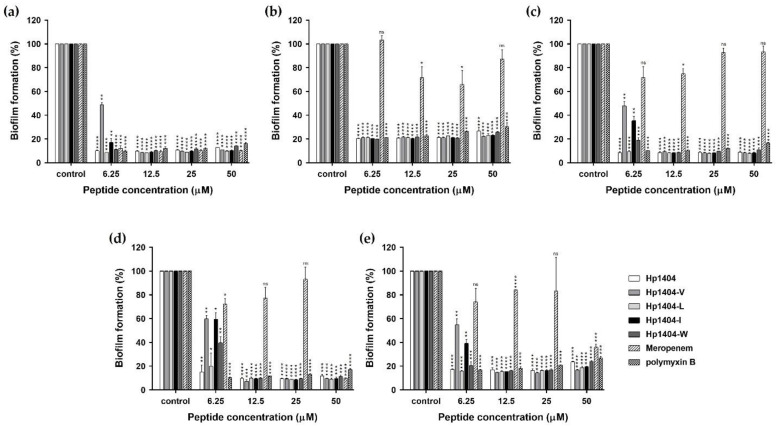
Anti-biofilm activity of peptides and antibiotics. The biofilm mass was measured in a crystal violet assay at wavelength 595 nm. (**a**) *A. baumannii* KCTC 2508, (**b**) *A. baumannii* 409081, (**c**) *A. baumannii* 719705 and (**d**) *A. baumannii* #3, and (**e**) *A. baumannii* #4. Data are presented as the mean ± SEM of three independent experiments. * *p* < 0.05, ** *p* < 0.01, *** *p* < 0.001, **** *p* < 0.0001 versus the control group. ns = Not statistically significant.

**Figure 8 ijms-22-05540-f008:**
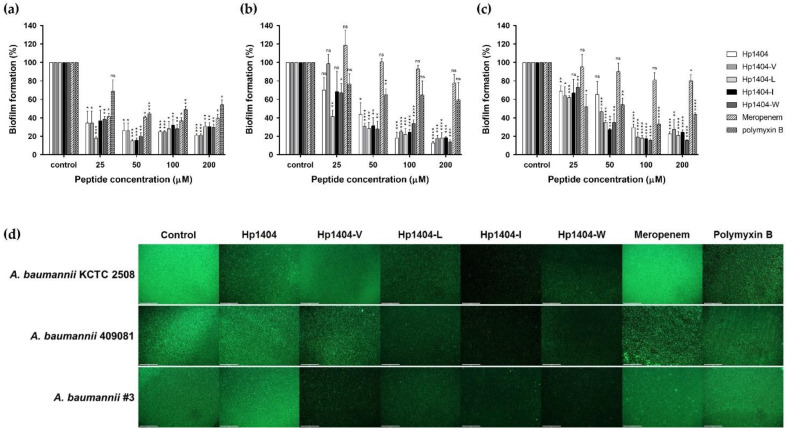
Biofilm reduction ability of peptides and antibiotics. The biofilm reduction rate was determined by crystal violet staining. (**a**) *A. baumannii* KCTC 2508, (**b**) *A. baumannii* #3 and (**c**) *A. baumannii* 409081. (**d**) Representative fluorescence images of *A. baumannii* strains stained with SYTO 9 (green) and evaluated using an EVOS™ FL Auto 2 Imaging System. Scale bar = 200 μm. Data are presented as the mean ± SEM of three independent experiments. * *p* < 0.05, ** *p* < 0.01, *** *p* < 0.001, **** *p* < 0.0001 versus the control group. ns = Not statistically significant.

**Figure 9 ijms-22-05540-f009:**
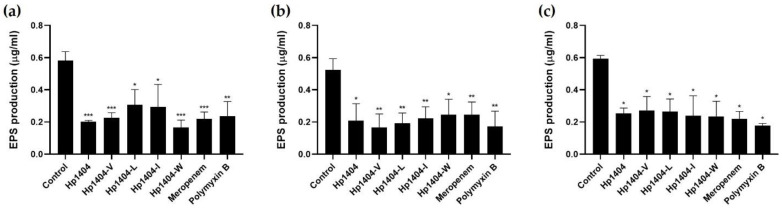
Effect of Hp1404 and its analog peptides on biofilm reduction and EPS production in *A. baumannii* strains. (**a**) *A. baumannii* KCTC 2508, (**b**) *A. baumannii* #3 and (**c**) *A. baumannii* 409081. Data are presented as the mean ± SEM of three independent experiments. * *p* < 0.05, ** *p* < 0.01, *** *p* < 0.001 versus the control group.

**Figure 10 ijms-22-05540-f010:**
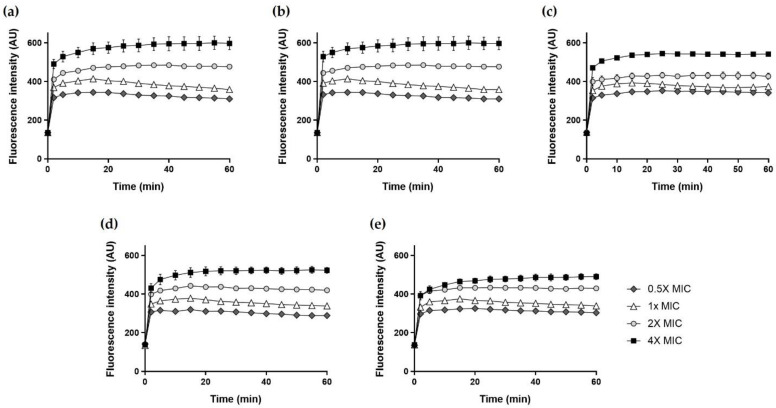
Outer membrane permeabilization of *A. baumannii* KCTC 2508 determined in NPN uptake assay. The increase of fluorescence intensity measured for 30 min at excitation wavelength of 350 nm and emission wavelength of 420 nm. (**a**) Hp1404, (**b**) Hp1404-V, (**c**) Hp1404-L (**d**) Hp1404-I and (**e**) Hp1404-W. Data are presented as the mean ± SEM of three independent experiments.

**Figure 11 ijms-22-05540-f011:**
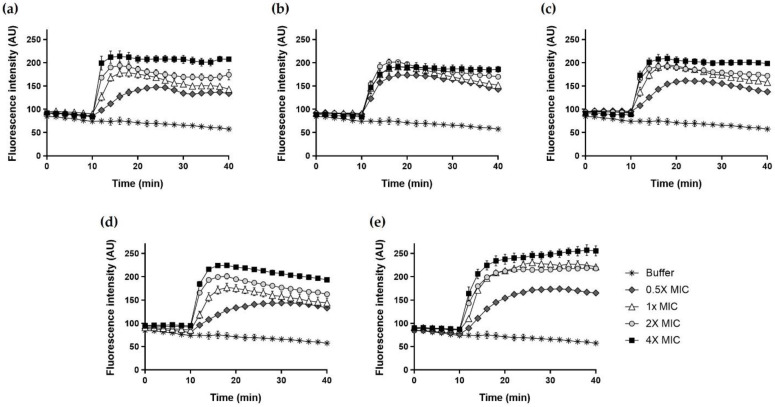
Cytoplasmic membrane depolarization of *A. baumannii* determined using membrane potential sensitive dye DisC_3_-5. The release of DisC_3_-5 was measured at an excitation wavelength of 622 nm and emission wavelength of 670 nm. (**a**) Hp1404, (**b**) Hp1404-V, (**c**) Hp1404-L (**d**) Hp1404-I and (**e**) Hp1404-W. Data are presented as the mean ± SEM of three independent experiments.

**Table 1 ijms-22-05540-t001:** Sequence and physicochemical properties of Hp1404 and its analog peptides.

Peptide	Sequence	R.T ^1^	Molecular Weight		Hydrophobicity (H) ^2^	Net Charge
			Observed	Calculated		
Hp1404	GILGKLWEGVKSIF-NH_2_	36.251	1545.9	1545.7	0.686	+1
Hp1404-A	GILGKLWEGVKSIA-NH_2_	30.719	1469.8	1469.1	0.580	+1
Hp1404-K	GILGKLWEGVKSIK-NH_2_	25.861	1526.9	1527.9	0.487	+2
Hp1404-V	GILGKLWEGVKSIV-NH_2_	33.166	1497.8	1497.0	0.645	+1
Hp1404-L	GILGKLWEGVKSIL-NH_2_	36.379	1511.9	1511.0	0.679	+1
Hp1404-I	GILGKLWEGVKSII-NH_2_	35.179	1511.9	1511.1	0.686	+1
Hp1404-W	GILGKLWEGVKSIW-NH_2_	33.889	1584.9	1584.2	0.719	+1

^1^ R.T (Retention Time): Mean retention time (min) in reversed-phase high-performance liquid chromatography. ^2^ Hydrophobicity (H) was calculated using HeliQuest site.

**Table 2 ijms-22-05540-t002:** Antimicrobial activity of peptides against gram-positive and gram-negative bacteria.

Microorganisms	MIC (µM)							
	Hp1404	Hp1404-A	Hp1404-K	Hp1404-V	Hp1404-L	Hp1404-I	Hp1404-W	Melittin
**Gram-positive**								
*S. aureus* ATCC 25923	3.13	12.5	25	6.25	1.56	3.13	3.13	0.78
*S. aureus* ATCC 29213	6.25	>25	>25	25	6.25	12.5	12.5	1.56
*L. monocytogenes* KCTC 3710	3.13	>25	>25	25	3.13	6.25	6.25	1.56
*B. cereus* KCTC 1012	12.5	>25	>25	25	12.5	25	25	3.13
**Gram-negative**								
*P. aeruginosa* ATCC 27853	12.5	>25	>25	>25	12.5	25	25	3.13
*E. coli* ATCC 25922	12.5	>25	>25	25	12.5	12.5	12.5	1.56
*K. pneumoniae* KCTC 2208	12.5	>25	>25	>25	12.5	25	12.5	1.56
*A. baumannii* KCTC 2508	3.13	25	25	6.25	3.13	3.13	3.13	1.56
*S. typhimurium* ATCC 14028	3.13	25	12.5	6.25	1.56	3.13	3.13	0.78

Data are presented as the mean ± SEM of three independent experiments using one-way ANOVA.

**Table 3 ijms-22-05540-t003:** Antimicrobial activity of peptides and antibiotics against *A. baumannii* strains isolated from patients.

Microorganisms	MIC (µM)							
	Hp1404	Hp1404-V	Hp1404-L	Hp1404-I	Hp1404-W	Melittin	Meropenem	Polymyxin B
*A. baumannii* 409081	3.13	6.25	3.13	3.13	3.13	1.56	200	0.78
*A. baumannii* 719705	3.13	12.5	3.13	6.25	3.13	1.56	100	0.78
*A. baumannii* 907233	3.13	6.25	3.13	3.13	3.13	1.56	200	0.78
*A. baumannii* #2	3.13	6.25	3.13	3.13	3.13	1.56	100	0.39
*A. baumannii* #3	3.13	6.25	3.13	3.13	3.13	1.56	50	0.78
*A. baumannii* #4	3.13	6.25	3.13	3.13	3.13	1.56	100	0.39

Data are presented as the mean ± SEM of three independent experiments using one-way ANOVA.

## Data Availability

Data is contained within the article.

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
