# Peer review of "Comparative Antimicrobial Activity of Hp404 Peptide and Its Analogs against Acinetobacter baumannii"

_ijms, 2021, doi:10.3390/ijms22115540_

Round 1

Reviewer 1 Report

Manuscript ID: ijms-1200950

Title: Antimicrobial, anti-biofilm activity and action mechanism of Hp1404 and its analog peptides against Acinetobacter baumannii

Antimicrobial resistance is one of the major threat to global health. Finding a suitable alternative against resistant strains is an unmet and urgent issue. The authors presented a nice topic, and the findings might somewhat help the community in the near future. However, there are some major flaws in the manuscript and several comments as below that need to be resolved to improve this manuscript to be considered for publication. The current status is not suitable for publication in IJMS.

  1. The title is too wordy with similar terminology. Compressed it. "A comparative antimicrobial study of Hp1404 peptide and its analogs against Acinetobacter baumannii" can be considered.
  2. The sentence stated between line 31-33 is required a citation.
  3. Line 36: write the sentence in other ways. Any drug can be resistant to invasive microbes in a particular condition.
  4. No specific objective has written in the introduction. Without stating the objective(s) of the study is not acceptable in any piece of writing.
  5. Line 64-70: This seems like a conclusion. The authors should remove these sentences from the introduction. Instead, state the clear problem statements and objectives of this study.
  6. The lack of coherence between paragraphs in the introduction is apparent. Authors need to establish coherence between paragraphs.
  7. Line 286-297: It seems like a problem statement. These sentences can be moved to the introduction section where appropriate.
  8. Results presentation is well, but the major flaws: authors did not show standard deviation/error in the table, for example, table 2,3, figure 3 etc., even though they mentioned the experiments had repeated thrice.
  9. Overall, the experimental section is well organized, but the authors did not mention how they extracted/got the peptide Hp1404 and how they produced its analog after designing it. If they purchased, they should mention it too.
  10. No statistical analysis appeared in the methods section even though the authors concluded their designed analogs were better than Hp1404. Without comparing the results with statistical tools, how it can be compared? The authors need to analyze all the findings and compare the results between Hp1404 and analogs, and among the analogs to find the best analogue against abumannii.
  11. Overall, the discussion is of too poor quality. The majority are like either introduction or redundancy of results. Authors should focus on the discussion of their findings in this section. They should compare their results with existing findings already reported by other researchers. It could be less, similar or higher effective than others findings. Any status should further explain the reason or mechanism. However, the authors mentioned in the title the molecular actions, but nothing discussed the molecular actions of analogue peptides against the studied bacterium.

Author Response

Response to reviewer’s

Editorial Board

IJMS

To the editor

Thank you for your valuable time and efforts to review our manuscript. We appreciated the opportunity to resubmit this manuscript. We found that the reviewer’s comments are very constructive and insightful and that the helpful criticism has enabled us to make a better and more informative manuscript. We have made most of the suggested alternations according to the reviewer’s comments. Below we have provided a detailed response to each comment, including:

Our response: red color and changed reference no(green color)

Altered text appeared in the manuscript- red color.

We look forward to receiving your final decision on this manuscript.

Sincerely

Park Yoonkyung, Ph.D.

Professor and Director

Department of Biomedical Science

College of Natural Science

Chosun University

375 Seosuk-Dong, Dong-Ku, Gwang-ju,

501-759, Korea.

Tel: 82-62-230-6854 (off), Fax: 82-62-225-6758

  1. mail: y_k_park@chosun.ac.kr

Comments and Suggestions for Authors

Manuscript ID: ijms-1200950

Title: Antimicrobial, anti-biofilm activity and action mechanism of Hp1404 and its analog peptides against Acinetobacter baumannii

Modified Title: Comparative antimicrobial activity of Hp404 peptide and its analogs against Acinetobacter baumannii

Antimicrobial resistance is one of the major threat to global health. Finding a suitable alternative against resistant strains is an unmet and urgent issue. The authors presented a nice topic, and the findings might somewhat help the community in the near future. However, there are some major flaws in the manuscript and several comments as below that need to be resolved to improve this manuscript to be considered for publication. The current status is not suitable for publication in IJMS.

1.The title is too wordy with similar terminology. Compressed it. "A comparative antimicrobial study of Hp1404 peptide and its analogs against Acinetobacter baumannii" can be considered.

Reply: Thank you for this comment. I have revised it as follows:

Comparative antimicrobial activity of Hp404 peptide and its analogs against Acinetobacter baumannii

2.The sentence stated between line 31-33 is required a citation.

Reply: Thank you for this comment. I have revised the text as follows.

(Modified text)

The World Health Organization recently cited antibacterial resistance as one of the three most important problems facing human health and reported that the number of multidrug-resistant (MDR) A. baumannii infections has increased rapidly, posing a public health threat worldwide [3-6].

(References)

  1. Kengkla, K.; Kongpakwattana, K.; Saokaew, S.; Apisarnthanarak, A.; Chaiyakunapruk, N. Comparative efficacy and safety of treatment options for MDR and XDR Acinetobacter baumannii infections: a systematic review and network meta-analysis. Journal of Antimicrobial Chemotherapy 2018, 73, 22-32.
  2. Mendelson, M.; Matsoso, M.P. The World Health Organization global action plan for antimicrobial resistance. SAMJ: South African Medical Journal 2015, 105, 325-325.
  3. World Health Organization. (‎2012)‎. The evolving threat of antimicrobial resistance: options for action: executive summary. World Health Organization. https://apps.who.int/iris/handle/10665/75389
  4. World Health Organization. (2015) Worldwide country situation analysis: response to antimicrobial resistance. Publication date: 29 April 2015; Number of pages: 42.
    ISBN: 978 92 4 156494 6; WHO reference number: WHO/HSE/PED/AIP/2015.1

3.Line 36: write the sentence in other ways. Any drug can be resistant to invasive microbes in a particular condition.

Reply: We agree with the reviewer’s comment; the text has been modified as follows. (Corrected the text in lines 51~53)

(Modified text)

Therefore, the development of new antimicrobial drugs against these multidrug-resistant pathogens is needed; antimicrobial peptides (AMPs) are considered an alternative to antibiotics.

4.No specific objective has written in the introduction. Without stating the objective(s) of the study is not acceptable in any piece of writing.

5.Line 64-70: This seems like a conclusion. The authors should remove these sentences from the introduction. Instead, state the clear problem statements and objectives of this study.

Reply: We agree with the reviewer’s comments 4 and 5. The sentences in lines 64–70 appeared as a conclusion; thus, we have removed the sentences and added the purpose of the study.

(Modified text)

Hence, the purpose of this study has been to design a peptide less toxic than its parent Hp1404 peptide and to investigate the antibacterial activity of the newly designed antimicrobial peptide on multidrug-resistant strains. We designed analog peptides by replacing the position 14 amino acid with different amino acids. Antimicrobial activity against gram-positive, gram-negative, and multidrug-resistant bacteria of the designed analog peptides was tested, and their mechanism of action was examined. Furthermore, we investigated the activity of peptides against A. baumannii strains and their ability to inhibit and eliminate bacterial biofilms. Our findings suggest that the newly designed peptides have the potential to be promising candidates as antimicrobial agents against A. baumannii strains.

6.The lack of coherence between paragraphs in the introduction is apparent. Authors need to establish coherence between paragraphs.

Reply: Thank you for this comment. I have revised the text as follows.

(Modified text)

Acinetobacter baumannii is a rod-shaped gram-negative bacterium and among the major causes of opportunistic nosocomial infections, known as ESKAPE pathogens (Enterococcus faecium, Staphylococcus aureus, Klebsiella pneumoniae, A. baumannii, Pseudomonas aeruginosa, and Enterobacter species). ESKAPE pathogens are highly antibiotic-resistant and difficult to treat, accounting for the majority of nosocomial and community-acquired infections and show high mortality rates [1,2]. The World Health Organization recently cited antibacterial resistance as one of the three mostimportant problems facing human health and reported that the number of multidrug-resistant (MDR) A. baumannii infections has increased rapidly, posing a public health threat worldwide [3–6]. To treat A. baumannii infection, antibiotics such as β-lactams or carbapenems with a broad spectrum are used as first-line options, but strains that have recently gained carbapenem resistance have emerged. Polymyxin or tigecycline is used to treat carbapenem-resistant A. baumannii [7], but most strains are multidrug resistant, and thus treatment options are very limited [8]. Multidrug-resistant A. baumannii causes opportunistic infections, including septicemia, pneumonia, endocarditis, skin and wound infections, and urinary tract infections following hospitalization of patients with severe illness [9,10]. This increased bacterial resistance to antibiotics is attributed to A. baumannii’s ability to form biofilms.

Biofilms are communities of microorganisms attached to a solid surface and produce a matrix known as extracellular polymeric substances (EPS) [11]. EPS are composed mostly of polysaccharides, glycoproteins, glycolipids, and extracellular DNA [12-15]. The matrix acts to protect bacteria, enabling them to survive in difficult-to-grow environments. Therefore, enhanced resistance or resistance to antibiotics and other antimicrobial agents com-pared to plankton bacteria is a typical feature of biofilms. Most bacterial infections are thought to be related to bacterial biofilms, and many cases of device-related infections have been reported. Device-related infections typically occur in medical devices such as catheters, contact lenses, mechanical heart valves, and artificial joints [16]. Therefore, the development of new antimicrobial drugs against these multidrug-resistant pathogens is needed; antimicrobial peptides (AMPs) are considered an alternative to antibiotics.

Organisms such as humans, animals, and plants are exposed to various microbial at-tacks; they protect themselves from these threats by producing antimicrobial peptides (AMPs) via the innate immune system [17,18]. AMPs, which have cationic and amphipathic properties, consist of fewer than 50 amino acids and show potent activity against bacteria, fungi, yeast, and viruses [19-21]. AMPs also exert anticancer activities and regulate inflammatory responses and wound healing [21-24]. Killing of bacteria leads to membrane destruction by causing the formation of pores or micelles and inhibition of protein synthesis or DNA replication by binding to RNA and DNA [25]. Despite the outstanding properties of AMPs, they exhibit disadvantages such as hemolytic activity against red blood cells and sensitivity to salt. Since antimicrobial peptides have the disadvantage of being sensitive to a salt environment, it is important to develop peptides that maintain their antibacterial activity in the presence of salts [26]. AMPs that can overcome these shortcomings would be promising candidates for treating bacterial infectious diseases [27].

(References)

  1. Kengkla, K.; Kongpakwattana, K.; Saokaew, S.; Apisarnthanarak, A.; Chaiyakunapruk, N. Comparative efficacy and safety of treatment options for MDR and XDR Acinetobacter baumannii infections: a systematic review and network meta-analysis. Journal of Antimicrobial Chemotherapy 2018, 73, 22-32.
  2. Mendelson, M.; Matsoso, M.P. The World Health Organization global action plan for antimicrobial resistance. SAMJ: South African Medical Journal 2015, 105, 325-325.
  3. World Health Organization. (‎2012)‎. The evolving threat of antimicrobial resistance: options for action: executive summary. World Health Organization. https://apps.who.int/iris/handle/10665/75389.
  4. World Health Organization. (2015) Worldwide country situation analysis: response to antimicrobial resistance. Publication date: 29 April 2015; Number of pages: 42.
    ISBN: 978 92 4 156494 6; WHO reference number: WHO/HSE/PED/AIP/2015.1.
  5. Brandl, K.; Plitas, G.; Mihu, C.N.; Ubeda, C.; Jia, T.; Fleisher, M.; Schnabl, B.; DeMatteo, R.P.; Pamer, E.G. Vancomycin-resistant enterococci exploit antibiotic-induced innate immune deficits. Nature 2008, 455, 804-807, doi:10.1038/nature07250.
  6. Han, H.M.; Ko, S.; Cheong, M.J.; Bang, J.K.; Seo, C.H.; Luchian, T.; Park, Y. Myxinidin2 and myxinidin3 suppress inflammatory responses through STAT3 and MAPKs to promote wound healing. Oncotarget 2017, 8, 87582-87597, doi:10.18632/oncotarget.20908.

  1. Line 286-297: It seems like a problem statement. These sentences can be moved to the introduction section where appropriate.

Reply: We agree with the reviewer’s comment and have added the information to the Introduction section.

(Modified text)

Acinetobacter baumannii is a rod-shaped gram-negative bacterium and among the major causes of opportunistic nosocomial infections, known as ESKAPE pathogens (Enterococcus faecium, Staphylococcus aureus, Klebsiella pneumoniae, A. baumannii, Pseudomonas aeruginosa, and Enterobacter species). ESKAPE pathogens are highly antibiotic-resistant and difficult to treat, accounting for the majority of nosocomial and community-acquired infections and show high mortality rates [1, 2]. In fact, the World Health Organization recently cited antibacterial resistance as one of the three most important problems facing human health, and reported that the number of multidrug-resistant (MDR) A. baumannii infections has increased rapidly, posing a public health threat worldwide [3-6]. To treat A. baumannii infection, broad-spectrum antibiotics such as β-lactams or carbapenems are used as first-line treatment options, but strains that have recently gained carbapenem resistance have emerged. Polymyxin or tigecycline is used to treat carbapenem-resistant A. baumannii infection [7], but most strains are multidrug resistant, and thus, treatment options are very limited [8]. Multidrug-resistant A. baumannii causes opportunistic infections, including septicemia, pneumonia, endocarditis, skin and wound infections, and urinary tract infections following hospitalization of patients with severe illness [9, 10]. This increased bacterial resistance to antibiotics is attributed to A. baumannii’s ability to form biofilms.

8.Results presentation is well, but the major flaws: authors did not show standard deviation/error in the table, for example, table 2,3, figure 3 etc., even though they mentioned the experiments had repeated thrice.

Reply: We agree with the reviewer’s comment. The error in the data has been indicated, and the legends have been revised.

(Modified text)

Data are presented as the mean ± SEM of three independent experiments using one-way ANOVA. * P < 0.05, ** P < 0.01.

9.Overall, the experimental section is well organized, but the authors did not mention how they extracted/got the peptide Hp1404 and how they produced its analog after designing it. If they purchased, they should mention it too.

Reply: Thank you for this comment. The text has been changed as follows.

(Modified text)

4.4. Peptide synthesis

The peptides were synthesized using the solid-phase-9-fluorenylmethoxycarbonyl (Fmoc) method, as reported previously [46], on a Rink amide 4-methylbenzhydrylamine resin using a Liberty microwave peptide synthesizer (CEM, Matthews, NY, USA). The following chemicals were used as linkage reagents: 0.45 M 2-(1H-benzotriazole-1-yil)-1,1,3,3-tetramethyluronium hexafluorophosphate (HBTU) diluted in dimethylformamide (DMF), 0.1 M N-hydroxybenzotriazole (HOBt) diluted in piperidine/DMF, and 2 M N, N-diisopropylethylamine (DIEA) diluted in N-methylpyrrolidone (NMP). After washing with dichloromethane (DCM), cleavage was performed by incubating for 2 h at 25 °C in a trifluoroacetic acid (TFA) solution containing water, phenol, and triisopropylsilane. The crude peptide was precipitated by dilution with ice-cold diethyl ether, and then spread on the tube wall and dried. After resuspension in water at 25 °C, the peptide was purified by reversed-phase high-performance liquid chromatography (RP-HPLC) on a Jupiter C18 column (4.6 × 250 mm, 300 Å, 5 µm; Phenomenex, Torrance, CA, USA). The molecular weights of the peptides were confirmed using matrix-assisted laser desorption ionization-time of flight (MALDI-TOF) mass spectrometry (Kratos Analytical Inc., Chestnut Ridge, NY, USA). The peptides were dissolved in deionized water (DI H2O) and stored at -20 °C. In this study, all the peptides used were > 95% pure.

(Reference)

  1. Fields, G.B.; Noble, R.L. Solid phase peptide synthesis utilizing 9‐fluorenylmethoxycarbonyl amino acids. International journal of peptide and protein research 1990, 35, 161-214.

10.No statistical analysis appeared in the methods section even though the authors concluded their designed analogs were better than Hp1404. Without comparing the results with statistical tools, how it can be compared? The authors need to analyze all the findings and compare the results between Hp1404 and analogs, and among the analogs to find the best analogue against a bumannii.

Reply: Thank you for this comment. The text has been revised as follows.

(Modified text)

4.18. Statistical analysis

All experiments were performed in triplicates in three independent experiments. Data are presented as the mean ± SEM. Statistical analysis of the results was performed with two-tailed Student’s t-test and one-way analysis of variance using GraphPad Prism version 8. 0 (La Jolla, CA, USA). The results were considered significant at *P < 0.05, ** P < 0.01, *** P < 0.001, ****P < 0.0001.

11.Overall, the discussion is of too poor quality. The majority are like either introduction or redundancy of results. Authors should focus on the discussion of their findings in this section. They should compare their results with existing findings already reported by other researchers. It could be less, similar or higher effective than others findings. Any status should further explain the reason or mechanism. However, the authors mentioned in the title the molecular actions, but nothing discussed the molecular actions of analogue peptides against the studied bacterium.

Reply: Thank you for this comment. As suggested, we have added the mechanism analysis to the Discussion section.

(Modified text)

The venom of various animal species is the source of biologically active molecules with attractive candidates for the development of novel therapeutics [29]. Antimicrobial peptides (AMPs) are bioactive molecules and notable sources of known antibacterial activity found in scorpions, ants, bees, snakes, wasps, and spiders. AMPs isolated from scorpions have been reported to effectively kill multidrug-resistant pathogens and possess anticancer activity but are slightly toxic [30]. Modifying AMPs could improve the antibacterial effect while reducing cytotoxicity, thereby overcoming the barriers preventing their development as therapeutic agents. Thus, new peptide analogs of Heterometrus petersii scorpion AMPs have been designed and synthesized, and their antimicrobial spectrum and hemolytic/cytotoxic activity have been determined.

We examined the antimicrobial activity of the parent peptide Hp1404 and its six analog peptides against gram-positive and gram-negative bacteria. Hp1404 and four analog pep-tides (Hp1404-W, Hp1404-L, Hp1404-I, and Hp1404-W) displayed antimicrobial activity, with excellent activity against A. baumannii, including clinical strains resistant to antibiotics. Particularly, the peptides substituted with Lue and Trp maintained antibacterial activity. Hp1404 and melittin were toxic towards mammalian cells, but the toxicity of the analog peptides decreased and cell viability increased by 80% even at a concentration 4-fold the active concentration, and the analogs exhibited stable antibacterial activity in a high-salt environment. Hp1404-L maintained antibacterial activity but showed toxicity similar to that of the parent Hp1404 peptide, while Hp1404-W showed no toxicity. For use as therapeutic agents, peptides need to have low toxicity and the ability to maintain the activity in a salt environment [37]. Cations such as Na+ interfere with the electrostatic interaction between the peptide and bacterial membrane, thereby reducing antimicrobial activity. Therefore, we determined whether each peptide maintains antibacterial activity in the presence of Na+; Hp1404-L and Hp1404-W showed excellent activity along with the parent peptide Hp1404. Thus, the analog peptides show potential for treating A. baumannii infections and overcome the limitations of Hp1404.

AMPs display random coil structures in their native forms; in bacterial membrane-mimicking environments, they display α-helix, β-sheet, or loop structures which act on the bacterial membrane [38]. It is known that AMPs such as melittin [39], magainin2 [40], cecropins [41], and LL37 [42] adopt α-helix structures. Hp1404 has been reported to form amphipathic α-helix when adsorbed to the bilayer lipid membrane of the bacterial membrane [28]. The structures of the analog peptides were confirmed under different conditions by CD spectroscopy. As a result, the parent peptide and its analog peptides formed a random coil in aqueous solution, and an α-helical structure was adopted in membrane-mimicking environments. The high amphiphilic and strong helix properties of the peptides are associated with their antimicrobial activity in high-salt environments (Figure 5). The helix structure and positive charges are strongly bound to negatively charged phospholipid bacterial membranes and efficiently exert their effects on them. We investigated the mechanism of action of outer membrane permeability and cytoplasmic membrane depolarization using NPN and DisC3-5. Peptides showed an increased fluorescence intensity of NPN and DiSC3-5 at each active concentration. This is consistent with the findings of studies on the mechanisms of action of AMP such as LL37 and melittin that act by destroying bacterial membranes. These results suggest that the peptides interact with the bacterial membrane to exert antibacterial effects by causing outer and inner membrane permeability in A. baumannii. Therefore, we observed that owing to this mechanism of action, analog peptides killed the bacteria in short time and thus showed potential as therapeutic agents.

References

  1. Primon-Barros, M.; José Macedo, A. Animal Venom Peptides: Potential for New Antimicrobial Agents. Curr Top Med Chem 2017, 17, 1119-1156, doi:10.2174/1568026616666160930151242.
  2. Harrison, P.L.; Abdel-Rahman, M.A.; Miller, K.; Strong, P.N. Antimicrobial peptides from scorpion venoms. Toxicon 2014, 88, 115-137.
  3. Zhu, X.; Dong, N.; Wang, Z.; Ma, Z.; Zhang, L.; Ma, Q.; Shan, A. Design of imperfectly amphipathic α-helical antimicrobial peptides with enhanced cell selectivity. Acta biomaterialia 2014, 10, 244-257.
  4. Dempsey, C.E. The actions of melittin on membranes. Biochimica et Biophysica Acta (BBA)-Reviews on Biomembranes 1990, 1031, 143-161.
  5. Kim, M.K.; Kang, N.H.; Ko, S.J.; Park, J.; Park, E.; Shin, D.W.; Kim, S.H.; Lee, S.A.; Lee, J.I.; Lee, S.H. Antibacterial and antibiofilm activity and mode of action of magainin 2 against drug-resistant Acinetobacter baumannii. International journal of molecular sciences 2018, 19, 3041.
  6. Steiner, H. Secondary structure of the cecropins: antibacterial peptides from the moth Hyalophora cecropia. FEBS letters 1982, 137, 283-287.
  7. Henzler Wildman, K.A.; Lee, D.-K.; Ramamoorthy, A. Mechanism of lipid bilayer disruption by the human antimicrobial peptide, LL-37. Biochemistry 2003, 42, 6545-6558.

Reviewer 2 Report

The manuscript is written in good English. The experiments, methods, and results are clearly described. The conclusions are supported by the results. In general the paper looks like a significant contribution in development of new antimicrobials. 

I have only a few minor comments for the authors:

It would be good to mention in the introduction that usually the salt environment have a negative impact on AMPs and it is necessary to study the antibacterial activity in presence of salts.

In the lines 295-297 the authors say "AMPs, as new antibacterial agents that kill antibiotic-resistant bacteria without causing resistance, are considered as alternatives to antibiotics." Why don't they cause the resistance? This statement looks to general and should be corrected or explained in the text. Indeed, "a low risk of bacterial resistance development" is not equal to "without causing resistance". It should be clarified.

A brief discussion about possible applications of the developed AMPs would make the paper more interesting for a broad audience. For example, recently it was shown that bacteriophages can be used to resensitize multidrug-resistant A. baumannii to antimicrobials (https://www.nature.com/articles/s41564-020-00830-7). Is it possible to use the AMPs to reach the same effect? 

Author Response

Response to reviewer’s

Editorial Board

IJMS

To the editor

Thank you for your valuable time and efforts to review our manuscript. We appreciated the opportunity to resubmit this manuscript. We found that the reviewer’s comments are very constructive and insightful and that the helpful criticism has enabled us to make a better and more informative manuscript. We have made most of the suggested alternations according to the reviewer’s comments. Below we have provided a detailed response to each comment, including:

Our response: red color

Altered text appeared in the manuscript- red color.

We look forward to receiving your final decision on this manuscript.

Sincerely

Park Yoonkyung, Ph.D.

Professor and Director

Department of Biomedical Science

College of Natural Science

Chosun University

375 Seosuk-Dong, Dong-Ku, Gwang-ju,

501-759, Korea.

Tel: 82-62-230-6854 (off), Fax: 82-62-225-6758

  1. mail: y_k_park@chosun.ac.kr

Comments and Suggestions for Authors

The manuscript is written in good English. The experiments, methods, and results are clearly described. The conclusions are supported by the results. In general, the paper looks like a significant contribution in development of new antimicrobials.

I have only a few minor comments for the authors:

It would be good to mention in the introduction that usually the salt environment has a negative impact on AMPs and it is necessary to study the antibacterial activity in presence of salts.

Reply: Thank you for this comment; it has been revised it as follows.

(Modified text)

Despite the outstanding properties of AMPs, they exhibit disadvantages such as hemolytic activity against red blood cells and sensitivity to salt. Since antimicrobial peptides have the disadvantage of being sensitive to a salt environment, it is important to develop peptides that maintain their antibacterial activity in the presence of salts [26]. AMPs that can overcome these shortcomings would be promising candidates for treating bacterial infectious diseases [27].

(References)

  1. Han, H.M.; Ko, S.; Cheong, M.J.; Bang, J.K.; Seo, C.H.; Luchian, T.; Park, Y. Myxinidin2 and myxinidin3 suppress inflammatory responses through STAT3 and MAPKs to promote wound healing. Oncotarget 2017, 8, 87582-87597, doi:10.18632/oncotarget.20908.

In the lines 295-297 the authors say "AMPs, as new antibacterial agents that kill antibiotic-resistant bacteria without causing resistance, are considered as alternatives to antibiotics." Why don't they cause the resistance? This statement looks to general and should be corrected or explained in the text. Indeed, "a low risk of bacterial resistance development" is not equal to "without causing resistance". It should be clarified.

Reply: We agree with the reviewer’s comment. The text has been modified as follows.

(Modified text)

Therefore, the development of new antimicrobial drugs against these multidrug-resistant pathogens is needed; antimicrobial peptides (AMPs) are considered an alternative to antibiotics.

A brief discussion about possible applications of the developed AMPs would make the paper more interesting for a broad audience. For example, recently it was shown that bacteriophages can be used to resensitize multidrug-resistant A. baumannii to antimicrobials (https://www.nature.com/articles/s41564-020-00830-7). Is it possible to use the AMPs to reach the same effect?

Reply: Thank you for this comment. The use of bacteriophages against resistant bacteria is very interesting. In case of AMPs, it can be applied because it has a synergistic effect with antibiotics. However, since the peptide studied in this paper exhibits excellent activity even at low concentrations, it represents the possibility of a good therapeutic agent without co-treatment.

Round 2

Reviewer 1 Report

The authors have taken the necessary steps to improve the standard of the manuscript based on the given comments. I have no further comment.